# Effects of individual variation and seasonal vaccination on disease risks

William S. Hart [1] ✉, Jina Amin [1], Hyeongki Park[2,3], Kosaku Kitagawa[3], Yong Dam Jeong [3,4], Alexander R. Kaye[5,6], Shingo Iwami [3,7,8,9,10,11] & Robin N. Thompson [1]

Estimates of the risk of a large outbreak resulting from pathogen introduction into a population are valuable for planning interventions. Two key factors affecting outbreak risks are variation in transmission between individuals (e.g., superspreading individuals) and change over time (e.g., through seasonality or changing population immunity due to vaccination). Here, we develop an outbreak risk estimation framework that accounts for both features simultaneously. To demonstrate the real-world application of our framework, we consider the design of annual COVID-19 booster vaccination campaigns, using a multi-scale approach incorporating an individual-level model of vaccine-induced antibody dynamics. Near the start of annual vaccine distribution, when population immunity is low, a high outbreak risk is possible; this can be mitigated by distributing vaccines over a longer period. We show that longer distribution periods are particularly beneficial if vaccine coverage and/or effectiveness is high, and if seasonality in transmission is limited.

Following the introduction of a pathogen into a host population, a key question for public health policy makers is whether a large outbreak is likely to occur or whether the pathogen will fade out after only a few infections[1]. To this end, mathematical models are widely used to estimate the outbreak risk: the probability of a single incident infection resulting in a large outbreak[1]. For example, the risk of sustained transmission in new locations was analysed following the initial emergence of the SARS-CoV-2 virus in China[2-5]. Subsequent studies considered the risk of local COVID-19 outbreaks[6-9], including assessing the effectiveness of public health measures such as mass testing[6] and vaccination[7].

A key transmission feature of many pathogens[10,11] including SARS-CoV-2[12-15] and Ebola virus[16] is substantial heterogeneity in transmission between different infected individuals. Because of variation in virus shedding[6,17] and/or contact patterns[13], a relatively small proportion of infected individuals may generate a large proportion of transmissions (superspreading; Supplementary Fig. 1). A seminal study by Lloyd-Smith et al.[11] showed how such individual variation can be included in outbreak risk estimates. For a specified reproduction number, the outbreak risk is typically lower when the extent of heterogeneity in infectiousness is greater, since (for example) a randomly chosen initial infected individual is then more likely to generate no transmissions at all (thus causing the outbreak to fade out immediately)[11]. However, when heterogeneity is greater, outbreaks that do occur are often more explosive, since superspreading events can cause rapid outbreak growth[11].

Another important feature considered in studies estimating outbreak risks is temporal variation in transmission[7,18-21], for example, arising due to seasonality[18,21] or the rollout of vaccines over time[7]. In the presence of such temporal variation, the naïve approach of estimating

[1]Wolfson Centre for Mathematical Biology, Mathematical Institute, University of Oxford, Oxford, UK. [2]School of Biomedical Convergence Engineering, Pusan National University, Busan, South Korea. [3]interdisciplinary Biology Laboratory (iBLab), Division of Natural Science, Graduate School of Science, Nagoya University, Nagoya, Japan. [4]Department of Mathematics, College of Natural Sciences, Pusan National University, Busan, South Korea. [5]Mathematics Institute, University of Warwick, Coventry, UK. [6]Zeeman Institute for Systems Biology and Infectious Disease Epidemiology Research (SBIDER), University of Warwick, Coventry, UK. [7]Institute of Mathematics for Industry, Kyushu University, Fukuoka, Japan. [8]Institute for the Advanced Study of Human Biology (ASHBi), Kyoto University, Kyoto, Japan. [9]Interdisciplinary Theoretical and Mathematical Sciences Program (iTHEMS), RIKEN, Saitama, Japan. [10]NEXT-Ganken Program, Japanese Foundation for Cancer Research (JFCR), Tokyo, Japan. [11]Science Groove Inc., Fukuoka, Japan. ✉e-mail: william.hart@maths.ox.ac.uk

the outbreak risk based on instantaneous transmission conditions (neglecting changes that occur following pathogen introduction) can lead to inaccurate outbreak risk estimates[18]. Accurate outbreak risk quantification, therefore, requires rigorous approaches that account for changes in transmission soon after the pathogen arrives in the host population. However, individual heterogeneity in transmission has not previously (to our knowledge) been considered alongside temporal variability when calculating the outbreak risk.

Here, we develop a general framework for calculating outbreak risks under time-dependent and heterogeneous transmission. Specifically, we consider a generalised renewal equation model that accounts for heterogeneous individual infectiousness, and derive equations satisfied by the outbreak risk. This framework is intended to be widely applicable, and we start by considering a simple illustrative example of its use to calculate seasonal outbreak risks. To demonstrate the application of our approach in a specific setting, we then consider the risk of localised COVID-19 outbreaks as a case study. Substantial SARS-CoV-2 transmission continues to occur worldwide, driven by viral evolution[22] and waning immunity[23–25]. Seasonal booster vaccinations are therefore offered in many countries, particularly to individuals at a higher clinical risk of severe infection outcomes[26]. We consider the COVID-19 outbreak risk under seasonal transmission[27,28] and annual booster vaccination, utilising a multi-scale approach that

incorporates an individual-level model of antibody dynamics following vaccination[29–31]. We focus on how the timing of annual vaccination distribution can be optimised to minimise the outbreak risk, finding that booster vaccine coverage and effectiveness, as well as the extent of seasonality in transmission, are key determinants of the optimal time of year (and duration of the year) for vaccines to be deployed. This exemplifies the practical conclusions that can be obtained from our general and extensible framework for outbreak risk quantification.

## Results

### Outbreak risks with time-dependent and heterogeneous transmission

We have developed a framework for calculating the time-dependent outbreak risk (the probability that a large outbreak results from the introduction of a single newly infected individual into the population on a given date), accounting for both temporal variation in transmission and heterogeneity in infectiousness between infected individuals. In Fig. 1, we demonstrate our method for a simple illustrative example.

Our method uses three inputs (Fig. 1A–C). First, time-dependent transmission is characterised by the instantaneous reproduction number, $R_t$ (Fig. 1A; here, $R_t$ is a periodic function of time, $t$, but other possibilities could be considered in our general framework). Second, similar to the approach of ref. 11, the extent of heterogeneity in

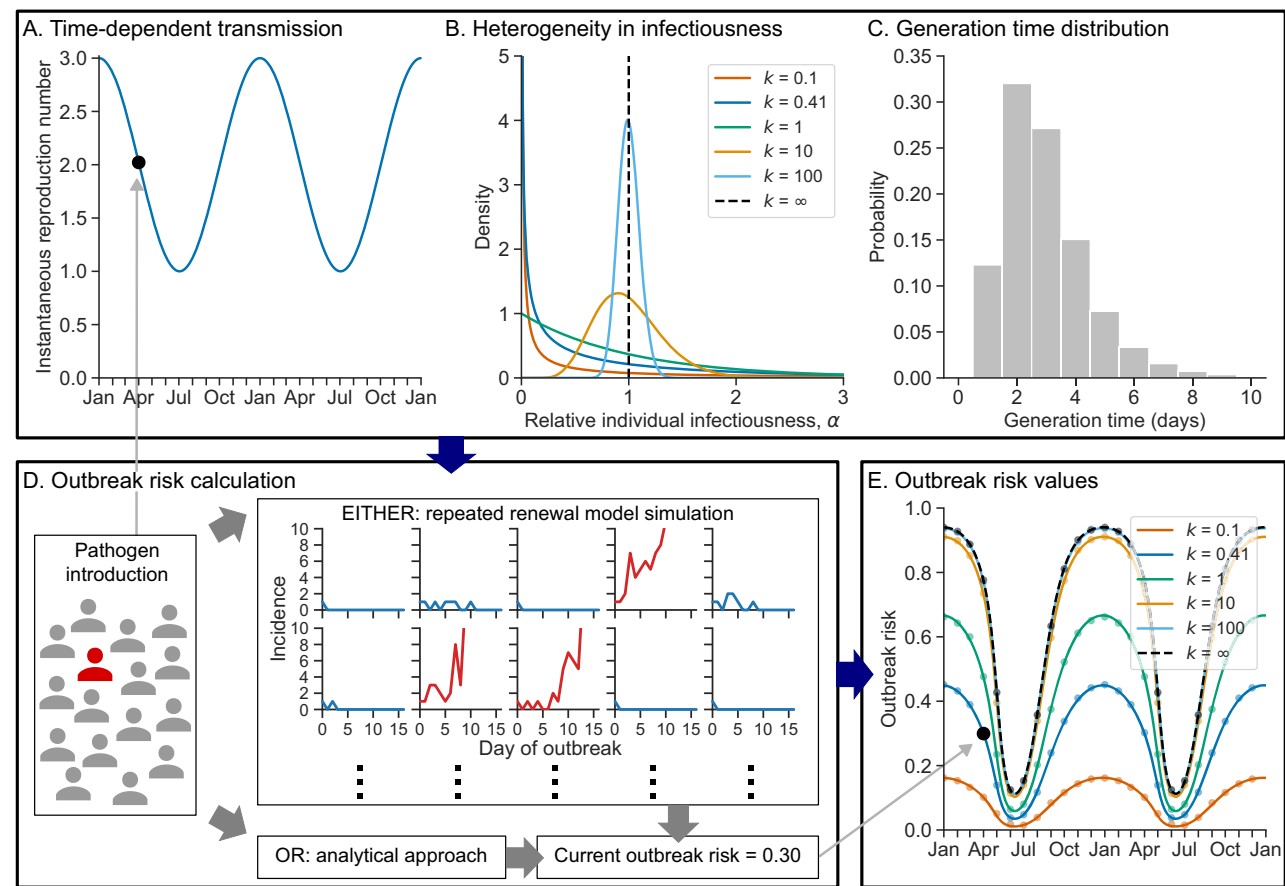

**Fig. 1 | Outbreak risks with time-dependent and heterogeneous transmission. A–C** Inputs to the outbreak risk calculation. **A** Instantaneous reproduction number, $R_t$, varying periodically over calendar time, $t$. **B** Heterogeneity in infectiousness, as characterised by a gamma distribution of relative individual infectiousness factors with mean 1 and dispersion (shape) parameter $k$ (with a lower value of $k$ corresponding to a greater degree of heterogeneity). This distribution is shown for a range of values of $k$ ($k = 0.1$: red; $k = 0.41$: blue; $k = 1$: green; $k = 10$: orange; $k = 100$: light blue; $k = \infty$: black dashed). **C** Generation time distribution[32]. **D** Illustration of how the outbreak risk at a given calendar time can be calculated either through

repeated stochastic simulation of a generalised renewal equation model (in this case, the outbreak risk is defined to be the proportion of simulations in which the daily incidence of new infections ever exceeds a specified threshold, here taken to be 30 daily infections; red curves show example simulations in which incidence exceeds the threshold, and blue curves show simulations in which the threshold is never exceeded), or by solving a system of analytically derived equations numerically. **E** Values of the outbreak risk, $p_t$, for the inputs shown in (**A–C**) (considering the same values of $k$ as in (**B**), with the same colour key), for both the analytical approach (lines) and simulation-based approach (dots).

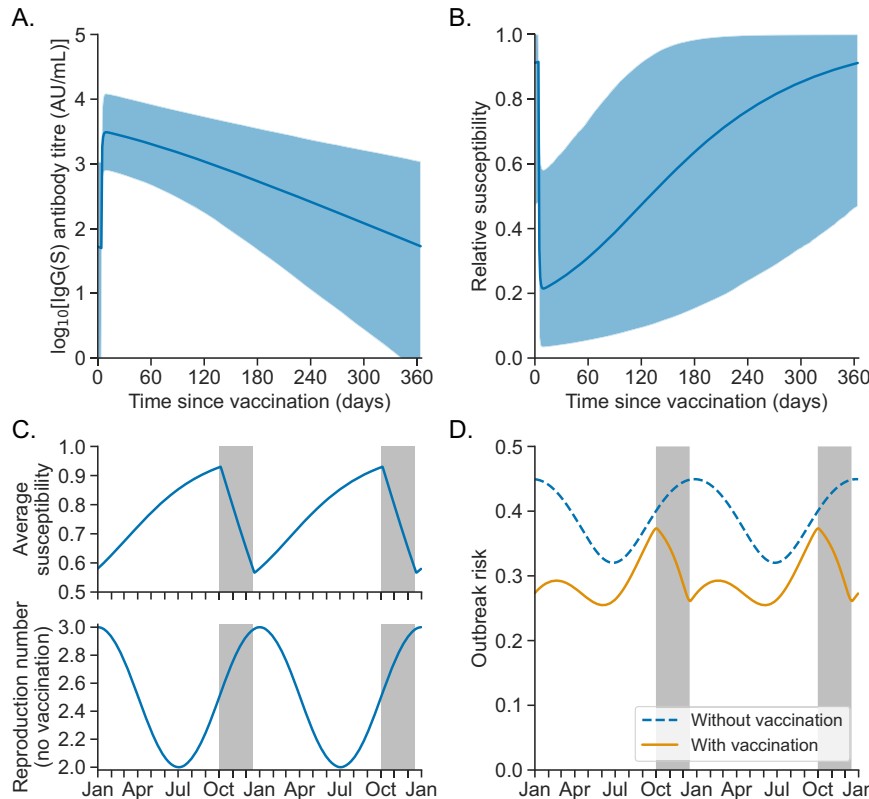

**Fig. 2 | Multi-scale modelling of the COVID-19 outbreak risk under annual booster vaccination. A** Log IgG(S) antibody titre relative to time since (most recent) vaccine dose. The curve shows the mean log titre (calculated over model-simulated antibody dynamics profiles for a synthetic cohort of 10,000 individuals), and the shaded region represents pointwise 95% prediction intervals (PIs) for individual antibody titres. **B** Relative susceptibility to SARS-CoV-2 infection following vaccination (compared to unvaccinated individuals, who are assumed to have relative susceptibility 1), which was assumed to be a function of individual antibody titre. The curve shows the mean susceptibility, and the shaded region represents pointwise 95% PIs for individual susceptibility. **C** Top: population-average susceptibility, $\eta_t$, as a function of calendar time, $t$, assuming 60% of the population receives booster vaccines between 1 October and 15 December each year (this time period is shown in the grey shaded regions). Bottom: instantaneous reproduction number, $R_{0,t}$, in the absence of booster vaccination. The instantaneous reproduction number when booster vaccination takes place, $R_t = \eta_t R_{0,t}$, is shown in Supplementary Fig. 2. **D** Outbreak risk, $p_t$, following the introduction of a single newly infected individual on day $t$, both without (blue dashed curve) and with (orange curve) booster vaccination, calculated using our analytical approach.

transmission between individuals is characterised by the dispersion parameter, $k$, which is the shape parameter of a gamma distribution of relative individual infectiousness factors—an infected individual with infectiousness factor $\alpha$ is expected (depending on stochasticity in the transmission process) to generate $\alpha$ times as many transmissions as a typical individual infected on the same day. A lower value of $k$ corresponds to a greater degree of heterogeneity between individuals (Fig. 1B and Supplementary Fig. 1). This formulation results in a negative binomial distribution for the number of transmissions (offspring) generated by an infected host, with dispersion parameter $k$[11]. The individual infectiousness factor distribution is shown for a range of values of $k$ in Fig. 1B ($k = 0.41$ is the pooled estimate for COVID-19 obtained in a previous meta-analysis[15]). Finally, the generation time distribution (the distribution of intervals between infection dates of infector-infectee transmission pairs) is also required as an input to our framework. The generation time distribution used throughout our analyses, which is a discretised version of a continuous-time estimate for the omicron SARS-CoV-2 variant[32], is shown in Fig. 1C.

Based on the specified inputs, the outbreak risk can be estimated via repeated simulation of a generalised renewal equation model (accounting for heterogeneity in infectiousness), calculated as the proportion of simulations in which the daily incidence of new infections ever exceeds a specified threshold (Fig. 1D). Alternatively, a system of equations satisfied by the outbreak risk at different time points can be derived analytically under the same transmission model and then solved numerically.

In Fig. 1E, the time-dependent outbreak risk (calculated using either the simulation-based or analytical approach) is shown for the inputs in Fig. 1A–C, again considering a range of values of the dispersion parameter, $k$. The simulation and analytical approaches give very similar outbreak risk values; therefore, only the analytical approach is considered in the remainder of our analyses (however, we note that the simulation-based approach may be needed in scenarios where $R_t < 1$ for much of the year—see Discussion). Similar to ref. 11 (in which temporal variation in transmission was not considered), we found that the outbreak risk on any given date is lower when there is a greater extent of heterogeneity in infectiousness (i.e., for smaller values of $k$).

## COVID-19 outbreak risk under annual booster vaccination

To show how our outbreak risk estimation approach can be applied in a more complex scenario than the general example shown in Fig. 1, we then considered temporal variation in the risk of localised COVID-19 outbreaks, and the impact of annual booster vaccination campaigns for mitigating this risk (Fig. 2). In our baseline analysis, we considered parameter choices representative of a hypothetical infectious novel SARS-CoV-2 variant and a vulnerable population with relatively high vaccine uptake (sensitivity analyses to model inputs are described later). We took a multi-scale approach, using a previously parameterised[31] individual-level antibody dynamics model[29,30] (Fig. 2A) to infer temporal variation in relative susceptibility to infection (compared to an individual who does not receive booster vaccines) following booster vaccination (Fig. 2B). This enabled us to estimate the

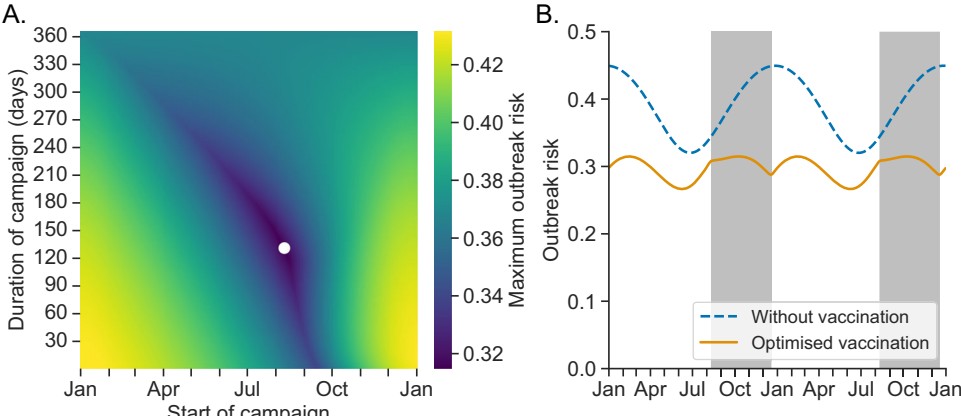

**Fig. 3 | Optimisation of the timing of booster vaccination campaigns. A** Annual maximum outbreak risk, plotted against the start date and duration of annual booster vaccine distribution (assuming 60% of the population is vaccinated each year). The white dot indicates the start date and duration at which the annual maximum outbreak risk is minimised. **B** Outbreak risk, $p_t$, without booster vaccination (blue dashed curve) and with optimised booster vaccine distribution timing (orange curve; the optimal distribution period each year is shown in the grey shaded regions).

population-averaged susceptibility at each calendar time, assuming in our baseline analysis that 60% of individuals (randomly chosen in the population) receive a vaccine dose each autumn/winter (Fig. 2C, top panel), with an equal number of individuals vaccinated each day of the booster campaign (1 October to 15 December each year). We combined these susceptibility values with a profile of seasonal transmissibility in the absence of booster vaccination (Fig. 2C, bottom panel) to obtain the instantaneous reproduction number, $R_t$, under annual booster vaccination (Supplementary Fig. 2). We then estimated the time-dependent outbreak risk (using our analytical approach) both without and with booster vaccination (Fig. 2D), assuming $k = 0.41$[15] and the generation time distribution shown in Fig. 1C.

Under our default assumptions, we found that booster vaccination reduces the outbreak risk, but does not eliminate it entirely (Fig. 2D). For example, the outbreak risk on 1 January each year (the assumed date of maximum transmissibility) is reduced by 39% by booster vaccination (from 0.45 to 0.27). On the other hand, the peak outbreak risk under booster vaccination (0.37) occurs on 2 October (one day after the start of annual vaccination) and is only 7% lower than the outbreak risk on the same date with no booster vaccination (0.40) because of limited population immunity on that date (since most individuals included in the booster vaccination campaign have not yet received their dose for the year in question).

### Optimisation of the timing of booster vaccination campaigns

We then considered how our multi-scale approach can be used to inform the timing of annual vaccine distribution. Specifically, we calculated the annual maximum outbreak risk for different possible start dates and durations of vaccine distribution (Fig. 3A), again under our baseline assumption that vaccine doses are received by 60% of the population each year. We found that the annual peak outbreak risk is minimised to 0.31 under a distribution window of 10 August to 18 December (compared to annual maximum values of 0.45 with no booster vaccination and 0.37 under the default distribution timing)—an earlier start date and similar end date to the default window considered in Fig. 2 (1 October to 15 December). The earlier start date ensures a lower outbreak risk when immunity is lowest (Fig. 3B) than under the default distribution timing (Fig. 2D), at the cost of a slightly higher outbreak risk near the winter transmission peak (due to waning immunity in those vaccinated earlier in the booster vaccination campaign).

### Factors determining optimal vaccine deployment timing

We analysed the impacts of numerous transmission features, which may vary based on factors such as SARS-CoV-2 variant and population

characteristics, on outbreak risk estimates under booster vaccination and optimal vaccine distribution timing (Fig. 4 and Supplementary Figs. 3–5). While the extent of heterogeneity in infectiousness between individuals (Fig. 4A–C) and the (mean) level of transmissibility in the absence of booster vaccination (Supplementary Fig. 3) both substantially affect quantitative outbreak risk estimates, we found that the optimal vaccine distribution timing is identical (10 August to 18 December) for all values of these parameters considered. In other words, under a policy goal of minimising the peak outbreak risk through an annual vaccination campaign with a fixed number of doses, the practical decision of when to deploy the vaccine doses is not affected by either variant transmissibility or the extent of individual heterogeneity. This is likely because, for the range of parameter values considered, these quantities do not substantially affect the extent to which the outbreak risk varies between different times of year.

On the other hand, we found the proportion of the population who receive annual booster vaccines, $\theta$, to be an important determinant of the optimal distribution strategy (Fig. 4D–F). Under our baseline distribution timing scenario, because of waning immunity, the outbreak risk at the start of the annual vaccine distribution period is only slightly reduced when even a large proportion of individuals are vaccinated (Fig. 4D). The optimal vaccine distribution window is longer when more individuals are vaccinated, since avoiding low immunity at times of moderate transmission becomes relatively more important compared to increasing (already high) immunity levels at times of peak transmission (Fig. 4E, F). Similarly, we found the optimal distribution window to be longer in scenarios where the extent of seasonality in transmission is smaller (Supplementary Fig. 4) or vaccines confer greater protection against infection (Supplementary Fig. 5).

## Discussion

Estimates of the risk of infectious disease cases introduced into a population leading to a large outbreak are useful for planning pathogen surveillance and interventions. Building on previous work, we have developed a general outbreak risk estimation framework that accounts for two key details: time-dependent transmission[18], and heterogeneity in the transmission potential between infected individuals[11]. To demonstrate how our framework could be applied to inform public health policy, we considered the risk of localised COVID-19 outbreaks under seasonal transmission and booster vaccination, using a multi-scale approach that incorporates an individual-level antibody dynamics model[29–31]. Typically, vaccines are distributed ahead of an expected winter peak in transmission, pre-empting the time of year at which

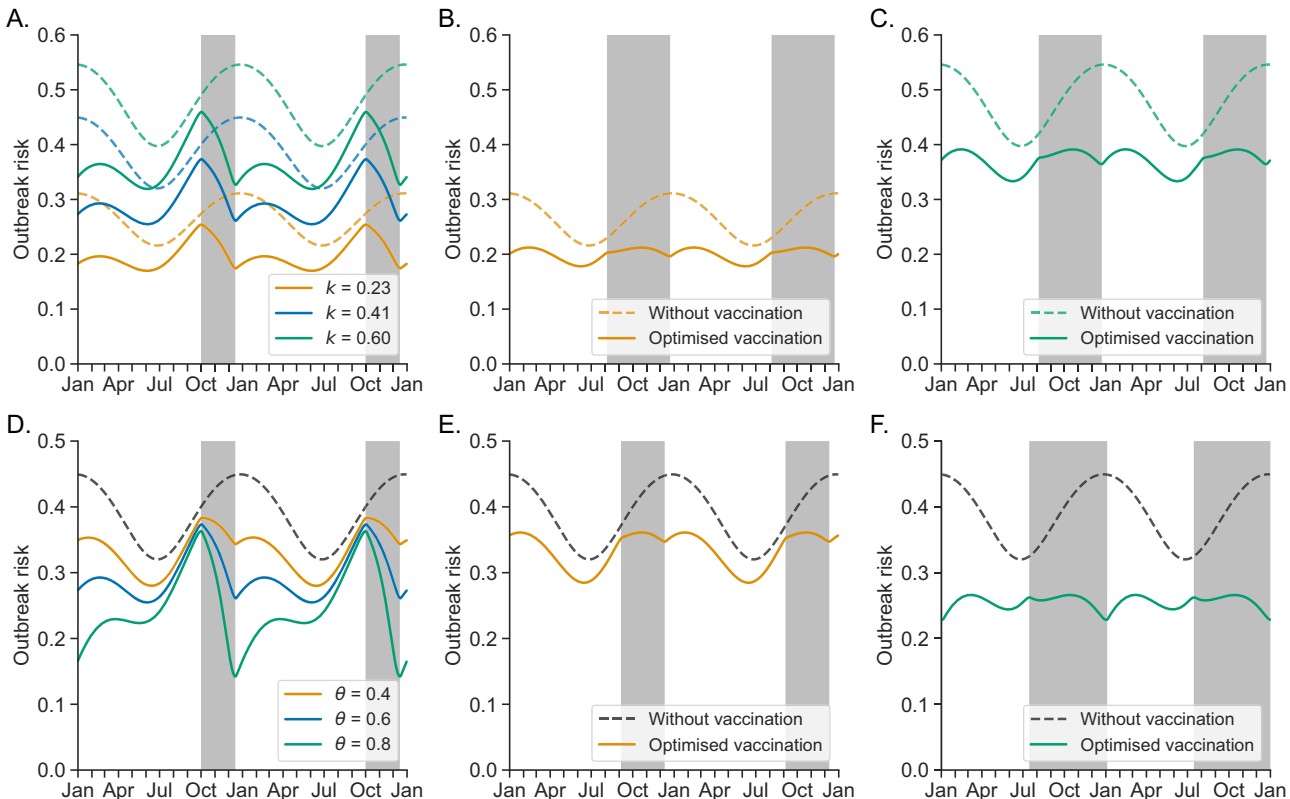

**Fig. 4 | Factors determining the impact of booster vaccination and the optimal timing of distribution. A–C** Effect of the dispersion parameter, *k*. **A** Outbreak risk without booster vaccination (dashed curves) and under the default assumed timing of booster vaccine distribution (solid curves), for values of *k* = 0.23 (orange), 0.41 (our default assumed value; blue) and 0.60 (green)—these values represent the lower 95% confidence interval bound, mean estimate and upper 95% confidence interval bound obtained for COVID-19 in ref. [15], respectively. **B, C** Outbreak risk without booster vaccination (dashed curves) and with optimised booster vaccine distribution timing (solid curves; the optimal distribution period each year is shown in the grey shaded regions) for *k* = 0.23 (**B**) and *k* = 0.60 (**C**). **D–F** Effect of the proportion of individuals vaccinated each year, *θ*. **D** Outbreak risk without booster vaccination (black dashed curve) and under the default assumed timing of booster vaccine distribution (solid curves), for values of *θ* = 0.4 (orange), 0.6 (our default assumed value; blue) and 0.8 (green). **E, F** Outbreak risk without booster vaccination (black dashed curves) and with optimised booster vaccine distribution timing (solid curves) for *θ* = 0.4 (**E**) and *θ* = 0.8 (**F**).

healthcare resources are under the highest strain[26]. Our analyses highlight that under a range of conditions, it may be beneficial to distribute vaccines over a relatively long period starting in late summer or early autumn (even aside from any operational constraints on the speed of distribution). This is because moderate autumn transmissibility, combined with waning immunity from the previous vaccination campaign, may otherwise generate a high outbreak risk near the start of annual vaccination.

In our baseline analysis, we considered parameter values representative of an infectious novel SARS-CoV-2 variant (with similar transmission characteristics to the omicron variant upon its initial emergence) in a population with relatively high booster vaccine uptake (as may be the case in a vulnerable population). However, even for COVID-19, the inputs to our outbreak risk estimation framework may vary depending on factors such as SARS-CoV-2 variant, vaccine availability and uptake, vaccine type(s) used, and characteristics of the local population. Therefore, we conducted several sensitivity analyses, finding that vaccine coverage and effectiveness, as well as the extent of seasonality in transmission, are important factors for determining optimal vaccination deployment timing. In general, a longer vaccine distribution window, starting earlier, decreases the outbreak risk near the time at which immunity is lowest (by bringing this time earlier in the year), at the expense of reduced immunity levels and so a higher outbreak risk around the winter transmission peak—the optimised distribution window considered here balances these factors to minimise the annual peak outbreak risk. If vaccine coverage is lower,

vaccine effectiveness is lower, and/or transmission seasonality is stronger, the optimal autumn vaccination campaign length is shorter, since the importance of maximising population immunity around the transmission peak is then increased.

Even in a scenario with higher vaccine uptake and optimised distribution timing, a non-zero outbreak risk may remain (Fig. 4F). While for simplicity we considered annual booster vaccination here, this finding suggests that more frequent vaccination may be required to prevent outbreaks of an infectious variant entirely in highly vulnerable populations such as care homes. In countries such as the UK, spring booster vaccinations have been offered to the most vulnerable members of the population, in addition to the annual autumn vaccinations. We expect that the optimal timing of autumn vaccination would be later and of shorter duration under twice- than once-yearly vaccination, since then the issue of low immunity near the start of vaccine distribution would be reduced.

In populations in which preventing outbreaks is impossible, the primary objective of booster vaccination campaigns would likely be to protect individuals from severe outcomes while the pathogen spreads. Decisions regarding which groups should receive booster vaccines, and how often, would then be based on individual risk of severe disease (considering waning immunity), alongside other factors such as economic cost. While we focussed on how the timing of annual booster vaccination (for a specified coverage level) can be optimised to reduce the outbreak risk, this is likely to correspond to a lower incidence of severe outcomes, since fewer outbreaks (particularly in vulnerable

populations) will usually lead to fewer severe cases (provided that the extent of individual heterogeneity remains unchanged).

We assumed that an optimal booster vaccine deployment strategy is one that minimises the annual peak of the outbreak risk (calculated each day assuming a single incident infection), but other possibilities could be considered. One option would be to consider the total annual outbreak risk, accounting for temporal variation in the likelihood of incident infections. If infectious introductions are most likely to occur when transmissibility (in the absence of vaccination) is highest, we expect that a shorter and later vaccination distribution window would be preferable to increase population immunity at that time. More generally, other modelling studies have explored how booster vaccination campaigns can be optimised, considering quantities such as peak hospitalisations[33], total deaths[34] and measures accounting for the extent of premature mortality[34]. Our approach here should be used alongside other sources of evidence (depending on policymakers' objectives).

While we considered COVID-19 as a case study, we expect both our outbreak risk estimation framework and insights about seasonal vaccination to be applicable more widely. We caution, however, that our analytical approach may not be suitable in scenarios in which transmission is impossible or unlikely at some times of year (for example, this is often the case for mosquito-borne diseases in temperate climates). Specifically, if outbreaks are always extinguished when transmission becomes low, even if a substantial number of cases have occurred already[35], the outbreak risk as calculated by numerically solving our analytically derived equations may be zero. In this scenario, we recommend using a simulation-based approach to estimate the outbreak risk (as used in Fig. 1E to verify the analytical outbreak risk estimates); careful consideration should be given to the specified incidence threshold for a major outbreak, since outbreak risk estimates may then be sensitive to the choice of threshold[35].

As with any modelling analysis, our results required us to make assumptions and simplifications. In each of our results, we were required to set parameter values characterising transmission features such as the mean transmission rate, the extent of seasonality, and the timing of peak transmission in the absence of vaccination (assuming transmission varies sinusoidally without vaccination), as well as vaccine coverage (which depends on both availability and uptake) and effectiveness. While these quantities are not known precisely, particularly for future SARS-CoV-2 variants, we ran a range of sensitivity analyses to draw general principles from our results, including identifying the parameters that affect the optimal time of year at which to deploy vaccines. If precise quantitative outbreak risk estimates are required, in principle a Bayesian approach could be adopted in which (expert-ascertained) uncertainty in future transmission conditions is incorporated into outbreak risk estimates and vaccination planning. We expect that uncertainty in the timing of peak transmission may lengthen the optimal duration of annual vaccine distribution to maintain population immunity over a range of possible times of high transmission.

We assumed that antibody dynamics only affect susceptibility to SARS-CoV-2 infection, but future work may extend our general framework to include a link between antibody titre (prior to breakthrough infection) and subsequent infectiousness. Specifically, our approach here could be combined with our previously developed framework for including individual-level virus dynamics in outbreak risk estimates[6], incorporating a relationship between prior immunity and virus dynamics during infection[36]. Because incorporating virus dynamics allows for detailed modelling of interventions such as antigen test screening[6], this extension would enable simultaneous modelling of vaccination and other interventions. Another target for future work is to relax our assumption of random vaccination by incorporating age structure; for COVID-19, older individuals have typically been prioritised for vaccination. Additionally, contact rates[34,37] and susceptibility[34] may vary between age groups, and these features could be included in an age-structured model. We note that non-random vaccination may alter the extent of individual heterogeneity in transmission—for example, targeting high-contact individuals would reduce heterogeneity. As a result, in addition to the outbreak risk, non-random vaccination strategies could affect the explosiveness of outbreaks that do occur (by altering the likelihood of superspreading); such impacts could be explored via model simulations.

Despite these simplifications, we have shown how outbreak risk estimates can be obtained in scenarios in which transmission is both heterogeneous between infected individuals and temporally varying. Applying our approach to the design of annual COVID-19 vaccination campaigns, we have identified vaccine coverage and effectiveness, and the extent of seasonality in transmission, as key determinants of the optimal vaccine distribution timing for minimising the annual peak outbreak risk. We expect these insights to be applicable to routine vaccination campaigns against other diseases (e.g., seasonal influenza) in addition to COVID-19. Our general modelling framework can be adapted to quantify outbreak risks in a wide range of different scenarios as an aid to public health policy-making.

## Methods
### Transmission model
We considered a discrete-time generalised renewal equation model that accounts for heterogeneity in infectiousness between different infected individuals. In the transmission model, each infected individual is assigned an individual infectiousness factor, $\alpha$, from a gamma distribution with shape parameter $k$ and mean one (although, in principle, other distributions could be used). If the reproduction number, $R$, remains constant during that individual's infection, the realised number of secondary infections generated by the individual is then assumed to follow a Poisson distribution with mean $\alpha R$. This Poisson-gamma formulation is equivalent to assuming that the realised number of secondary infections from a randomly chosen infected individual follows a negative binomial offspring distribution with mean $R$ and dispersion parameter $k$[11]. Smaller values of $k$ correspond to a greater degree of superspreading (Supplementary Fig. 1). The offspring distribution is a geometric distribution if $k = 1$, while a Poisson distribution is obtained in the limit $k \to \infty$[11].

More generally, accounting for temporal changes in the reproduction number, $R_t$, between different times, $t$, the model-simulated incidence of newly infected individuals, $I_t$, is generated as follows:

$$I_t \sim \text{Poisson}\left(\text{mean} = R_t \sum_{\tau=1}^{\infty} w_\tau J_{t-\tau}\right) \quad (1)$$

$$J_t \sim \text{Gamma}\left(\text{shape} = kI_t, \text{scale} = \frac{1}{k}\right) \quad (2)$$

Here, $w_\tau$ is the probability mass function of the discrete generation time distribution (the probability that the interval between the dates of infection of an infector-infectee pair is $\tau$ days), and $J_t = \sum_{i=1}^{I_t} \alpha_{t,i}$ represents the sum of the individual infectivity factors ($\alpha_{t,i}$) of the individuals infected on day $t$ (labelled $i = 1, \ldots, I_t$). We note that this formulation is valid because the sum of independent gamma-distributed random variables with identical scale parameters also follows a gamma distribution (with the same scale parameter).

### Outbreak risk
**Analytical approach.** The outbreak risk, $p_t$, is defined to be the probability (under the above transmission model, and assuming that the reproduction number is known at all times) that an outbreak, initiated with a single incident infection at time $t$ (i.e., taking $I_t = 1$, with $I_s = 0$ for $s < t$), does not become extinct. As shown in the

Supplementary Methods, the outbreak risk satisfies the following equation:

$$p_t = 1 - \left(1 + \frac{1}{k}\sum_{\tau=1}^{\infty} R_{t+\tau} w_\tau p_{t+\tau}\right)^{-k} \qquad (3)$$

In our numerical analyses, we assumed that $R_t$ is a periodic function of time (so that $p_t$ is also periodic with the same period). In that scenario, a closed system of equations for the values of $p_t$ (across all values of $t$), which can be solved numerically, can be obtained from the above expression (see the Supplementary Methods for details).

If $R_t = R$ is constant in time, then the implicit equation for the outbreak risk, $p_t = p$, derived in ref. 11 is recovered:

$$p = 1 - \left(1 + \frac{Rp}{k}\right)^{-k} \qquad (4)$$

The outbreak risk is then given by the largest solution less than one of Eq. (4)[11].

**Simulation-based approach.** An alternative method for calculating the outbreak risk at time $t$ involves simulating the renewal model a large number of times, initiating each simulation with a single incident infection at time $t$. The outbreak risk is then calculated as the proportion of model simulations in which the daily incidence of new infections ever exceeds a specified threshold. Each simulation is run until either the outbreak ends (i.e., there are no new infections for a period exceeding the maximal generation time) or the incidence threshold is exceeded. When we used the simulation-based approach in Fig. 1E, we conducted 20,000 model simulations for each time of pathogen introduction and used an incidence threshold of 30 infections.

## Preliminary analysis of seasonal transmission
We initially considered a simple scenario in which the reproduction number varies periodically with period 365 days:

$$R_t = 2 + \cos\left(\frac{2\pi t}{365}\right) \qquad (5)$$

Over the course of each year, $R_t$ therefore varies between one and three. We compared the outbreak risk, calculated using either the analytical or simulation-based approach, between different values of the dispersion parameter, $k$ (Fig. 1E), under the generation time distribution for COVID-19 described below and shown in Fig. 1C.

## COVID-19 booster vaccination case study
**Antibody dynamics model.** We considered a model of individual-level antibody dynamics following a dose of an mRNA COVID-19 vaccine[29–31]. In this model, the IgG(S) antibody titre, $A(\tau)$, of an individual at time since (most recent) vaccination $\tau$ evolves according to the ordinary differential equation,

$$\frac{dA}{d\tau} = \frac{HM(\tau)^m}{K^m + M(\tau)^m} - \mu A(\tau), \quad \tau \geq \tau_d \qquad (6)$$

where

$$M(\tau) = D e^{-\delta\tau} \qquad (7)$$

is the quantity of mRNA inoculated by the vaccine dose remaining. Here, $\tau_d$ represents a delay before the vaccine elicits an antibody response; for $0 \leq \tau < \tau_d$, the antibody titre of a previously vaccinated individual is assumed to continue to vary based on the remaining

inoculated mRNA and induced antibody titre from their previous vaccination dose (for a previously unvaccinated individual, the antibody titre is assumed to be zero until time $\tau_d$). Interpretations of the model parameters $H$, $m$, $K$, $\mu$, $D$ and $\delta$ are given in Supplementary Table 1. We note that we here slightly simplified the model compared to previous studies[29–31] by assuming that residual mRNA from previous vaccination doses is negligible (after the delay, $\tau_d$).

In a previous study[31], the antibody dynamics model was fitted to longitudinal data collected following a booster (third overall) dose of the BNT162b2 or mRNA-1273 vaccine. In that study[31], densely sampled blood data from 12 healthcare workers were considered first, indicating that only the parameters $H$ and $m$ exhibit substantial variability between individuals. Then, sparsely sampled data from a larger cohort of 1618 individuals were used to obtain individual estimates of $H$ and $m$, with remaining parameters fixed based on model fits to the healthcare worker data. Here, we used the mean and standard deviation of logarithms of individual parameter estimates for the 1618 individuals (Supplementary Table 1) to generate synthetic antibody profiles (see below and the Supplementary Methods), assuming each parameter value varies between individuals according to a lognormal distribution.

**Individual susceptibility model.** We assumed that individual relative susceptibility (i.e. one minus the level of vaccine protection against infection), $S(\tau)$, is given in terms of the antibody titre by

$$S(\tau) = \frac{1}{1 + \exp\left(\kappa \log_{10}\left(\frac{A(\tau)}{A_{1/2}}\right)\right)} \qquad (8)$$

This functional relationship has previously been applied to relate susceptibility to neutralising antibody titres[38–41], and the application of the same formula here is justified by the high correlation between neutralising antibodies and the IgG(S) antibody titre[31]. We assumed a value of $\kappa = 3.1$ as estimated in ref. 40. The parameter $A_{1/2}$, representing the antibody titre at which 50% protection is conferred, is likely to depend on both vaccine type and the SARS-CoV-2 variant under consideration[25,38,39,41]. Therefore, we took a default value of $A_{1/2} = 1,000$ AU/mL—this gave a maximum average vaccine protection against infection of 79% (Fig. 2B), which is comparable to previous model estimates of the effectiveness of a fourth vaccine dose against symptomatic disease due to the omicron SARS-CoV-2 variant[25]. We also considered alternative values of $A_{1/2}$ in Supplementary Fig. 5.

**Annual booster vaccination.** We suppose that booster vaccines are received by a proportion, $\theta$, of the population, each year between calendar times $t_s$ and $t_e$ (assuming that vaccinations are spread equally between each day in this time window).

To calculate the mean population susceptibility, $\eta_t$, on each calendar day $t$, we considered a synthetic cohort of 10,000 individuals. Considering individual variation in antibody dynamics model parameters and assuming an interval of one year between successive vaccine doses, we calculated periodic solutions of the antibody dynamics model for each individual. This allowed us to calculate the expected susceptibility, $\bar{S}(\tau)$, of individuals included in booster vaccination campaigns, as a function of time since most recent vaccine dose, $\tau$. Details of this calculation are given in the Supplementary Methods. Assuming individuals who do not receive booster vaccinations have relative susceptibility one, and for simplicity neglecting variation in relative susceptibility within each day, we then calculated

$$\eta_t = (1 - \theta) + \theta \sum_{t_v = t_s}^{t_e} \bar{S}(t - t_v) \qquad (9)$$

In our main analyses, we took $\theta = 0.6$ to represent a vulnerable population with relatively high booster uptake. Alternative values of $\theta$ are considered in Fig. 4D–F. Initially, we considered booster vaccination rollout between $t_s = 1$ October and $t_e = 15$ December, which is similar to the original planned timetable for 2023 in England[42] (the 2023 booster vaccination start date was later moved forwards in response to the emergence of the BA.2.86 omicron subvariant[43]). We then explored how the rollout timing could be optimised to minimise the annual peak outbreak risk.

**Seasonal transmissibility.** In the absence of booster vaccination (i.e., assuming all individuals are entirely susceptible), we assumed the instantaneous basic reproduction number to be given by[27,44]

$$R_t = R_{0,t} = \overline{R_0}\left(1 + \varepsilon \cos\left(\frac{2\pi t}{365}\right)\right) \qquad (10)$$

We assumed that the basic reproduction number is highest on 1 January each year, which is consistent with the respiratory infectious disease burden in the UK typically being highest in December and January[42].

We took default values of $\varepsilon = 0.2$, as previously assumed for COVID-19 based on estimates for other coronaviruses in ref. 27, and $\overline{R_0} = 2.5$, giving a maximum annual basic reproduction number ($R_{0,t} = 3$) consistent with estimates of the reproduction number of the omicron SARS-CoV-2 variant (BA.1 subvariant) shortly after it emerged in late 2021[45]. Different values of $\overline{R_0}$ and $\varepsilon$ are considered in Supplementary Figs. 3, 4, respectively.

The instantaneous reproduction number accounting for booster vaccination was then calculated as $R_t = \eta_t R_{0,t}$.

**Heterogeneity in infectiousness.** In our initial analysis, we assumed a dispersion parameter of $k = 0.41$, as estimated for COVID-19 in ref. 15. Alternative values of $k$ are considered in Fig. 4A–C.

**Generation time distribution.** Throughout our analyses, we assumed a lognormal continuous generation time distribution (i.e., distribution of the interval between the exact times of infection of an infector-infectee pair) with mean 3.0 days and standard deviation 1.5 days (this corresponds to lognormal distribution parameters, describing the mean and standard deviation of the natural logarithm of the continuous generation time, of $\mu = 0.98$ and $\sigma = 0.47$, respectively), as estimated for household SARS-CoV-2 omicron variant transmissions in ref. 32. We then discretised this distribution using the method described in refs. 46,47, truncating the discretised distribution at 14 days (after which the remaining probability mass is negligible). The discretised generation time distribution is shown in Fig. 1C.

### Reporting summary
Further information on research design is available in the Nature Portfolio Reporting Summary linked to this article.

## Data availability
All data are available at https://github.com/will-s-hart/covid-boosters (archived at https://doi.org/10.5281/zenodo.15800215). Individual parameter estimates for the antibody dynamics model were originally obtained by ref. 31.

## Code availability
All code used in the analyses is available at https://github.com/will-s-hart/covid-boosters (archived at https://doi.org/10.5281/zenodo.15800215). Computer code was written in Python (compatible with version 3.12).

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

## Acknowledgements

Thanks to members of the Wolfson Centre for Mathematical Biology at the University of Oxford, particularly the Infectious Disease Modelling group, for useful discussions about this work. This research was funded by the EPSRC through a Vacation Internship (to J.A., grant number EP/W524311/1). W.S.H., A.R.K. and R.N.T. acknowledge the support of the JUNIPER partnership (grant number MR/X018598/1). The collaboration between R.N.T. and S.I. was supported by a Royal Society International Exchange award (grant number IES-R3-193037). For the purpose of Open Access, the authors have applied a CC BY public copyright licence to any Author Accepted Manuscript (AAM) version arising from this submission.

## Author contributions

W.S.H.: conceptualisation, methodology, formal analysis, software, investigation, visualisation, supervision, writing (original draft) and writing (review and editing). J.A.: methodology, formal analysis, software, investigation, visualisation and writing (review and editing). H.P., K.K., Y.D.J. and A.R.K.: methodology and writing (review and editing). S.I.: conceptualisation, methodology and writing (review and editing). R.N.T.: methodology, supervision and writing (review and editing).

## Competing interests

The authors declare no competing interests.
