## [Transparent Peer Review file · Nature Communications]

Effects of individual variation and seasonal vaccination on disease risks

Corresponding Author: Dr William Hart

Version 0:

Reviewer comments:

Reviewer #1

(Remarks to the Author)

This paper develops a model to determine outbreak risk when including heterogeneity in susceptibility and seasonal variation. To demonstrate the utility of the model, an optimization for timing of seasonal booster vaccination is performed. Under the baseline model assumptions (60% receiving booster and other parameters taken from literature values related to COVID-19), the optimal vaccine timing and duration is a long period up to and through the rise in cases. The work is well-written and thorough, although most of the interesting derivations are relegated to the supplemental material. Some of the assumptions of the model are quite simplistic, such as seasonal variation being a cosine function without any variability of timing or magnitude of peaks, or that R_t is greater than one at all points in the year. It would improve the paper to have further discussion on how more realistic outbreaks structure may alter results.

Minor comments:

Line 116: "Similarly" should be "similar"

Line 136: The definition for the specific threshold is buried in the supplemental methods. It would be helpful include this information early or to point to where to find this information.

Line 172 (Figure 2): Outbreak risk is not clearly defined yet. It would be helpful include this information early or to point to where to find this information.

Line 227 (Figure 4): Theta is not defined yet. It would be helpful include this information early or to point to where to find this information.

(Remarks on code availability)

Reviewer #2

(Remarks to the Author)

In this manuscript, Hart and colleagues introduce a mathematical modelling framework to examine how (a) heterogeneity in individual-level infectiousness and (b) temporal variation in transmissibility (e.g., due to vaccine uptake and seasonality) impact the probability that an outbreak will occur when a pathogen is introduced into a community. They find that outbreaks are most likely when both individual-level infection heterogeneity and baseline transmissibility are high. They also find that, for a fixed level of vaccine uptake and for a seasonal virus, distributing vaccination over a longer period of time can reduce the risk of outbreaks occurring near the start of the virus' transmission season.

The work addresses an important and under-studied question. While the impact of individual-level infectiousness heterogeneity and of seasonality/vaccine uptake are well understood individually, I am not aware of other studies that have explicitly examined how these drivers combine to impact outbreak risk. While seasonality and individual infectiousness heterogeneity are not intervenable factors, vaccine uptake is, and the authors conduct a thoughtful analysis of how vaccination strategies can be optimized to reduce the risk of outbreaks. The mathematical analysis is robust and appropriate — the use of both simulations and theory (and specifically, the use of renewal equations) is especially appreciated and is a major strength of this manuscript.

I have three major comments for the authors' consideration:

1. Framing. While it's useful to have COVID-19 as a baseline example, I would encourage the authors to frame their work more generally. This might include moving the current first paragraph of the introduction (lines 44-49) to later in the manuscript. Also, some of the discussion about COVID-19 seemed outdated or inaccurate; for example, in lines 47-49, the statement that "the potential for localized outbreaks remains, including in vulnerable populations" doesn't seem to take into account the fact that major, year-round transmission of COVID-19 is still occurring across all age and risk groups!

2. Multiple introductions. The authors assess outbreak risk in terms of the probability that a single pathogen introduction yields an epidemic that exceeds some threshold size. However, in many cases, and certainly in the case of COVID-19, we might expect many introductions to occur. I don't think that the authors necessarily need to extend their analysis to account for this, but perhaps they could find examples in the literature that assess how outbreak risk varies with number/intensity of introductions and discuss how this might apply to their findings. I expect that, with multiple introductions, the space between the curves in Figure 1E might diminish, for example, with all of them shifting upwards.

3. Outbreak explosiveness. A key point in the article by Lloyd-Smith and colleagues (ref 20) is that heterogeneity in the secondary infection distribution not only reduces the risk of outbreaks, but also makes the resulting outbreaks potentially more explosive. As a result, I question whether reducing outbreak risk is the appropriate goal. In the Discussion (lines 297-299), the authors state that "fewer outbreaks (particularly in vulnerable populations) will usually lead to fewer severe cases." This isn't necessarily true — if we have fewer outbreaks, but the outbreaks that occur are more intense, then we haven't necessarily gained anything, and in fact we might be even worse off because it's harder to manage intense outbreaks than mild ones. Ideally, the authors would include a formal, mathematical assessment of outbreak size/intensity and determine how various vaccination strategies affect not just outbreak risk, but also outbreak intensity. However, I'm also reluctant to recommend a major set of new analyses for a paper that is already interesting and useful. At the very least, the authors should heavily caveat their findings with these points in mind, and outline how one might extend their work to account for outbreak explosiveness.

(Remarks on code availability)

The code is well-structured and is appropriate for reproducing the authors' findings.

Reviewer #3

(Remarks to the Author)

This paper investigates the impact of simultaneous variations in transmission between individuals (superspreaders) and over time (seasonality, population changes, vaccination) on disease outbreaks. This topic, of course, is highly relevant to policy-makers, and the paper directly addresses the implications for policy makers. As an example, the paper considers annual COVID-19 booster vaccine campaigns. Results show that there is a high risk of outbreak risk at the start of the vaccine booster campaign, and that longer distribution periods can mitigate this risk. The paper is also generally well-written. The booster vaccine example effectively highlighted the issues at hand, and there is a good discussion of the recommended public policies and limitations of the model. I have only a few points of clarification and comments to improve the paper detailed below.

Figure 3 shows that a longer duration of a vaccine booster campaign can result in a higher maximum outbreak risk. Is this true? And if so, why? Are there any trade-off in duration length that could be considered? Some insights with respect to this result are discussed briefly in the discussion: it's due to waning immunity and thus moderate transmission in the autumn. However, they should also be mentioned here and elaborated in the discussion. More specifically, the discussion in lines 278-281 could be expanded upon.

Lines 222-225 are unclear (it appears there are some grammatical errors and the sentence is a little long). Why is this result the case?

The authors mention spring boosters to vulnerable members of the population. Did they consider this in their analysis, and how might it affect their results?

(Remarks on code availability)

I am not sufficiently knowledgeable of the language to evaluate the code. However, the code is well commented and the README file has sufficient instructions.

Version 1:

Reviewer comments:

Reviewer #2

(Remarks to the Author)

The authors have sufficiently addressed my concerns.

(Remarks on code availability)

The code appears well documented and usable.

Response to reviews

Please note: In the responses below, whenever we refer to line numbers in the revised manuscript, these refer to the version of the manuscript with “track changes”. We have uploaded two versions of the revised manuscript – one with, and one without, track changes.

Reviewer #1 (Remarks to the Author):

This paper develops a model to determine outbreak risk when including heterogeneity in susceptibility and seasonal variation. To demonstrate the utility of the model, an optimization for timing of seasonal booster vaccination is performed. Under the baseline model assumptions (60% receiving booster and other parameters taken from literature values related to COVID-19), the optimal vaccine timing and duration is a long period up to and through the rise in cases. The work is well-written and thorough, although most of the interesting derivations are relegated to the supplemental material.

Response

We thank the reviewer for their comments, which have helped us to improve our manuscript. We decided to locate the more complex mathematical derivations underlying our methodology in the Supplementary Material to ensure that our work is understandable to the broadest possible audience, but we are pleased that the reviewer (like us!) finds these details interesting.

Some of the assumptions of the model are quite simplistic, such as seasonal variation being a cosine function without any variability of timing or magnitude of peaks, or that R_t is greater than one at all points in the year. It would improve the paper to have further discussion on how more realistic outbreaks structure may alter results.

Response

We have added discussion of these points as requested (lines 451-483). We hypothesise that uncertainty in the timing of peaks is likely to widen the optimal vaccination distribution window to maintain population immunity over the range of possible times of high transmission (lines 481-483).

In principle, our approach could be applied in some scenarios in which R_t is less than one at certain times of year. However, as we discuss in detail in the revised manuscript (lines 451-463), the simulation-based approach may be needed if sustained transmission is impossible for large portions of the year (for example, for mosquito-borne diseases in temperate climates). This is because the outbreak risk value obtained from our analytically derived equations may be zero if outbreaks are always extinguished when transmission becomes low, even if a substantial number of infections have occurred already. Investigating the optimal timing and duration of vaccination campaigns in such scenarios using model simulations is a target for future research.

Minor comments:

Line 116: "Similarly" should be "similar"

Response

We have corrected this in the revised manuscript (line 187).

Line 136: The definition for the specific threshold is buried in the supplemental methods. It would be helpful include this information early or to point to where to find this information.

Response

We have added this information to the caption to Figure 1 (lines 179-180).

Line 172 (Figure 2): Outbreak risk is not clearly defined yet. It would be helpful include this information early or to point to where to find this information.

Response

We have added this information to the caption to Figure 2 (lines 270-271).

Line 227 (Figure 4): Theta is not defined yet. It would be helpful include this information early or to point to where to find this information.

Response

We have added this information to the caption to Figure 4 (lines 342-343).

Reviewer #2 (Remarks to the Author):

In this manuscript, Hart and colleagues introduce a mathematical modelling framework to examine how (a) heterogeneity in individual-level infectiousness and (b) temporal variation in transmissibility (e.g., due to vaccine uptake and seasonality) impact the probability that an outbreak will occur when a pathogen is introduced into a community. They find that outbreaks are most likely when both individual-level infection heterogeneity and baseline transmissibility are high. They also find that, for a fixed level of vaccine uptake and for a seasonal virus, distributing vaccination over a longer period of time can reduce the risk of outbreaks occurring near the start of the virus' transmission season.

The work addresses an important and under-studied question. While the impact of individual-level infectiousness heterogeneity and of seasonality/vaccine uptake are well understood individually, I am not aware of other studies that have explicitly examined how these drivers combine to impact outbreak risk. While seasonality and individual infectiousness heterogeneity are not intervenable factors, vaccine uptake is, and the authors conduct a thoughtful analysis of how vaccination strategies can be optimized to reduce the risk of outbreaks. The mathematical analysis is robust

and appropriate — the use of both simulations and theory (and specifically, the use of renewal equations) is especially appreciated and is a major strength of this manuscript.

I have three major comments for the authors' consideration:

Response

We thank the reviewer for their comments, which have helped us to improve our manuscript.

1. Framing. While it's useful to have COVID-19 as a baseline example, I would encourage the authors to frame their work more generally. This might include moving the current first paragraph of the introduction (lines 44-49) to later in the manuscript. Also, some of the discussion about COVID-19 seemed outdated or inaccurate; for example, in lines 47-49, the statement that “the potential for localized outbreaks remains, including in vulnerable populations” doesn't seem to take into account the fact that major, year-round transmission of COVID-19 is still occurring across all age and risk groups!

Response

We have revised the Introduction of our manuscript to frame our work more generally and corrected discussion about COVID-19 as suggested (see in particular lines 74-85 and 133-138).

2. Multiple introductions. The authors assess outbreak risk in terms of the probability that a single pathogen introduction yields an epidemic that exceeds some threshold size. However, in many cases, and certainly in the case of COVID-19, we might expect many introductions to occur. I don't think that the authors necessarily need to extend their analysis to account for this, but perhaps they could find examples in the literature that assess how outbreak risk varies with number/intensity of introductions and discuss how this might apply to their findings. I expect that, with multiple introductions, the space between the curves in Figure 1E might diminish, for example, with all of them shifting upwards.

Response

In principle, it would be straightforward to account for multiple introductions in our approach, since chains of transmission due to different incident infections occur independently in renewal equation models. For example, if the outbreak risk following a single new infection on day t is p_t , then the outbreak risk following n independently acting new infections on day t is $1 - (1 - p_t)^n$ (see for example the review paper by Southall et al. – reference (1) of our manuscript). The reviewer is therefore correct that the outbreak risk would increase with the number of introductions (except in scenarios in which p_t is equal to zero or one). An important target for future work, which we discuss in lines 434-441 of the revised manuscript, would be to account for a time-dependent likelihood of introductions in our approach (potentially including the possibility of multiple introductions occurring on the same day). If introductions

are most likely to occur when transmissibility (in the absence of vaccination) is also highest, we expect that a shorter and later vaccination distribution window would be preferable to mitigate the high outbreak risk at such a time.

3. Outbreak explosiveness. A key point in the article by Lloyd-Smith and colleagues (ref 20) is that heterogeneity in the secondary infection distribution not only reduces the risk of outbreaks, but also makes the resulting outbreaks potentially more explosive. As a result, I question whether reducing outbreak risk is the appropriate goal. In the Discussion (lines 297-299), the authors state that “fewer outbreaks (particularly in vulnerable populations) will usually lead to fewer severe cases.” This isn’t necessarily true — if we have fewer outbreaks, but the outbreaks that occur are more intense, then we haven’t necessarily gained anything, and in fact we might be even worse off because it’s harder to manage intense outbreaks than mild ones. Ideally, the authors would include a formal, mathematical assessment of outbreak size/intensity and determine how various vaccination strategies affect not just outbreak risk, but also outbreak intensity. However, I’m also reluctant to recommend a major set of new analyses for a paper that is already interesting and useful. At the very least, the authors should heavily caveat their findings with these points in mind, and outline how one might extend their work to account for outbreak explosiveness.

Response

We thank the reviewer for raising these important points. As suggested by the reviewer, we have included appropriate caveats in the revised manuscript and described how our work could be extended to consider outbreak explosiveness. In particular, we have noted in the Introduction that outbreaks can be more explosive under greater heterogeneity (since superspreading events can cause rapid outbreak growth; lines 93-95), caveated our statement about fewer outbreaks leading to fewer severe cases (lines 429-433), and discussed outbreak explosiveness in the context of targeted vaccination strategies that affect the extent of heterogeneity (lines 492-510) – for example, vaccinating high-contact individuals would reduce heterogeneity. As we now discuss (lines 509-510), the impact of non-random vaccination on outbreak explosiveness could be explored in future work via (generalised) renewal model simulations (please note that Lloyd-Smith et al. also used simulations, rather than an analytic approach, to analyse outbreak explosiveness).

Reviewer #2 (Remarks on code availability):

The code is well-structured and is appropriate for reproducing the authors' findings.

Response

We are pleased the reviewer found our code to be well-structured and appropriate.

Reviewer #3 (Remarks to the Author):

This paper investigates the impact of simultaneous variations in transmission between individuals (superspreaders) and over time (seasonality, population changes, vaccination) on disease outbreaks. This topic, of course, is highly relevant to policy-makers, and the paper directly addresses the implications for policy makers. As an example, the paper considers annual COVID-19 booster vaccine campaigns. Results show that there is a high risk of outbreak risk at the start of the vaccine booster campaign, and that longer distribution periods can mitigate this risk. The paper is also generally well-written. The booster vaccine example effectively highlighted the issues at hand, and there is a good discussion of the recommended public policies and limitations of the model. I have only a few points of clarification and comments to improve the paper detailed below.

Response

We thank the reviewer for their comments, which have helped us to improve our manuscript.

Figure 3 shows that a longer duration of a vaccine booster campaign can result in a higher maximum outbreak risk. Is this true? And if so, why? Are there any trade-off in duration length that could be considered? Some insights with respect to this result are discussed briefly in the discussion: it's due to waning immunity and thus moderate transmission in the autumn. However, they should also be mentioned here and elaborated in the discussion. More specifically, the discussion in lines 278-281 could be expanded upon.

Response

We have expanded the relevant parts of the Results (lines 293-297) and Discussion (lines 398-406) as suggested. The choice of campaign length (and timing) indeed involves a trade-off: a shorter campaign, starting later, provides increased immunity at the winter transmission peak, but moves the time of minimum immunity (just before the start of annual vaccination) later in the year, so that transmission in the absence of vaccination is higher when immunity is at its lowest level. Therefore, the optimal vaccination strategy essentially balances the outbreak risk when immunity is lowest and the outbreak risk when transmission (without vaccination) is highest.

Lines 222-225 are unclear (it appears there are some grammatical errors and the sentence is a little long). Why is this result the case?

Response

We have rephrased this for clarity (lines 318-321). The lack of impact of mean transmissibility and the extent of individual heterogeneity on the optimal vaccination timing is likely because, while these quantities are important determinants of the outbreak risk, they have little effect on the extent by which the outbreak risk varies between different times of year (which is key for determining optimal vaccination timing; see lines 321-324).

The authors mention spring boosters to vulnerable members of the population. Did they consider this in their analysis, and how might it affect their results?

Response

For simplicity, we focused on annual booster vaccination in our work (see lines 408-409). As we note in our revised manuscript (lines 420-423), we expect that (as well as reducing the outbreak risk) twice-annual vaccination may lead to the optimal timing of autumn vaccination being later and of shorter duration, since then the problem of low immunity near the start of vaccine distribution (which leads to a benefit in starting to distribute vaccines earlier) would be reduced.

Reviewer #3 (Remarks on code availability):

I am not sufficiently knowledgeable of the language to evaluate the code. However, the code is well commented and the README file has sufficient instructions.

Response

We are pleased the reviewer found our code to be appropriately documented.